# Formulation of Tioconazole and *Melaleuca alternifolia* Essential Oil Pickering Emulsions for Onychomycosis Topical Treatment

**DOI:** 10.3390/molecules25235544

**Published:** 2020-11-26

**Authors:** Barbara Vörös-Horváth, Sourav Das, Ala’ Salem, Sándor Nagy, Andrea Böszörményi, Tamás Kőszegi, Szilárd Pál, Aleksandar Széchenyi

**Affiliations:** 1Institute of Pharmaceutical Technology and Biopharmacy, Faculty of Pharmacy, University of Pécs, H-7624 Pécs, Hungary; barbara.horvath@aok.pte.hu (B.V.-H.); ala.salem@aok.pte.hu (A.S.); sandor.nagy@aok.pte.hu (S.N.); 2Department of Laboratory Medicine, Medical School, University of Pécs, H-7624 Pécs, Hungary; pharma.souravdas@gmail.com (S.D.); koszegi.tamas@pte.hu (T.K.); 3János Szentágothai Research Center, University of Pécs, H-7624 Pécs, Hungary; 4Department of Pharmacognosy, Faculty of Pharmacy, Semmelweis University, H-1085 Budapest, Hungary; aboszormenyi@gmail.com

**Keywords:** pickering emulsions, onychomycosis topical treatment, tioconazole, tea tree essential oil, antifungal activity

## Abstract

Onychomycosis is a disease that affects many adults, whose treatment includes both oral and topical therapies with low cure rates. The topical therapy is less effective but causes fewer side effects. This is why the development of an effective, easy to apply formulation for topical treatment is of high importance. We have used a nanotechnological approach to formulate Pickering emulsions (PEs) with well-defined properties to achieve site-specific delivery for antifungal drug combination of tioconazole and *Melaleuca alternifolia* essential oil. Silica nanoparticles with tailored size and partially hydrophobic surface have been synthesized and used for the stabilization of PEs. In vitro diffusion studies have been performed to evaluate the drug delivery properties of PEs. Ethanolic solution (ES) and conventional emulsions (CE) have been used as reference drug formulations. The examination of the antifungal effect of PEs has been performed on *Candida albicans* and *Trichophyton rubrum* as main pathogens. In vitro microbiological experimental results suggest that PEs are better candidates for onychomycosis topical treatment than CE or ES of the examined drugs. The used drugs have shown a significant synergistic effect, and the combination with an effective drug delivery system can result in a promising drug form for the topical treatment of onychomycosis.

## 1. Introduction

Onychomycosis is a fungal infection of nails and nail bed and occurs on both finger and toenails. This fungal infection affects about 11.4% [1] of the adult population and is responsible for more than 50% of nail diseases [2]. For the treatment of onychomycosis, oral, topical, mechanical, and chemical therapies or a combination of these methods are used in practice [3]. The therapy is a long process (10–12 months) and has a poor cure rate [4]. Oral therapy is the most effective, but in the case of prolonged use, orally administered drugs can cause severe side effects because of their high toxicity [5]. Drug interactions can also occur, which is the main reason for the contraindication of oral therapy. In such cases, topical therapy is recommended, and it is a more attractive alternative for patients. Unfortunately, the topical treatment of onychomycosis is limited because the nail plate acts as a barrier to drug diffusion. Its hydrophilic nature and keratinized structure reduce the diffusion of high molecular weight or lipophilic antifungal drugs [6]. In order to enhance the penetration of drugs, diffusion enhancers (e.g., mechanical pretreatment [7], phosphoric acid [8], keratolytes [9]) or an appropriate formulation should be used [3].

The azole antifungal agents have been used since 1980 for topical and oral therapy of fungal infections, among others for onychomycosis; in clinical treatment, they are used for superficial and systemic fungal infections with safety [10]. The water solubility of generally applied antifungal azole derivatives in the treatment of fungal nail infections is very low (<0.01 mg/mL) [10]; therefore, their formulation contains organic solvents in most cases. Commercially available nail lacquers contain organic solvents to increase the solubility of antifungal drugs, but these solvents act unfavorably to the drug permeability [11]. Their restricted drug delivery ability is caused by the rapid evaporation of organic solvents, so some drugs remain on the surface of the nail [12]. Several researches have proved that drugs with aqueous-based formulations have higher nail permeability than non-aqueous ones [13,14]. Another problem with azole derivatives is that fungi can become resistant to the drug in a long-lasting treatment [15]. A combination of azoles with antifungal essential oil (EO) could solve this problem because fungi cannot easily acquire resistance to multiple antifungal components of EOs [16]. Moreover, the azole derivatives could show synergistic antifungal activity with some essential oils (EOs) [17], presumably because of their different mode of action. The azole derivatives inhibit the action of cytochrome P-450 enzyme, lanosterol demethylase of fungi [18], thereby preventing the synthesis of ergosterol, while the EOs damage the cell membranes and organelles of fungi [19]. Because of the lipophilic character of azole derivatives [10], it is likely that it can be dissolved in lipophilic EO, and their solution can be used for drug formulation.

The oil in water type emulsions as water-based drug formulations can provide a possible way to overcome the water solubility problems. Conventionally, the emulsions are stabilized with surfactants. In a long-lasting topical treatment, the use of surfactants should be avoided because they can cause irritation and side effects, or in some cases, they can get into the blood circulation [20]. Using particle-stabilized emulsions, i.e., Pickering emulsions (PEs) [21] instead of a conventional emulsion, has several advantages. The solid particles spontaneously adsorb on the oil-water interface and form a shell-like structure on the PE droplet surface [22]. This adsorption can be considered as irreversible adsorption because the solid particles have higher adsorption energy than surfactants on oil-water interfaces [23], so the stability of PEs can be the same or better than conventional emulsions. Another important parameter for emulsion-based drug formulation and therapy is the size of emulsion droplets. The droplet size of PE can be influenced by several parameters, such as emulsification time and energy, oil to solid particle volume ratio, and concentration of oil phase [24]. The fungal hypha damages the nail structure, creating pores in µm size range [25], with porosity in the 5–20% range depending on pretreatment of the nail [26]. An emulsion droplet with the appropriate size could penetrate into the porous nail structure and retain the antifungal drug on the nail bed for a longer period, which is the main site of reinfection [27], and thereby a targeted drug delivery can be achieved. Inert and biocompatible particles should be chosen as stabilizing particles in PEs drug formulation.

Silica nanoparticles (SNPs) are widespread in pharmaceutical technology in topical treatments because of their favorable chemical and surface properties, thermal stability, mechanical resistance, and biocompatibility [28,29]. The effects of topically applied SNPs have been examined in detail [30], and it has been found that they have no toxic effect even after prolonged usage. Because of the above-mentioned advantageous properties of SNPs, they can be suitable for PEs stabilization. The PEs are most stable when the partial wetting conditions of stabilizing particles are the same for the oil and water phase [31]. The native SNPs are hydrophilic because of the high number of free silanol groups at their surface [32], which has to be modified with organic ligands to achieve appropriate wettability and strong stabilizing effect.

In the present study, our aim was to formulate PE of an azole derivative and antifungal essential oil as an alternative formulation for onychomycosis topical treatment. Tioconazole (TIO) has a broad antifungal activity for common dermatophytes, which has proved to be efficient for the topical treatment of fungal infections [33]. Nenoff et al. determined that *Melaleuca alternifolia* (MA) EO (tea tree EO) inhibited the growth of several clinical fungal isolates, so they suggested its use in the topical treatment of fungal infections [34]. For PE formulation, we used the solution of TIO in MA EO. Synthesis and surface modification of SNPs with different sizes were performed, and they were used as stabilizing agents of PEs. We characterized the stability, droplet size, and emulsion type of PEs. In vitro diffusion studies were also performed through artificial membranes. The aim of the in vitro diffusion test was to compare drug delivery characteristics of different formulations in the membranes that have similar porous structure and surface properties as the nail plate and nail bed. The antifungal activity against *Candida albicans* and *Trichophyton rubrum*—the species mainly responsible for fungal nail infections—has been investigated [3].

## 2. Results and Discussion

### 2.1. Characterization and In Vitro Diffusion Study of PEs

#### 2.1.1. Characterization of SNPs

The size distribution and PDI values for synthesized and surface-modified SNPs were determined by DLS and TEM. Data for mean diameter and PDI values are presented in Table 1. The TEM images showed that SNPs were monodispersed, spherical, and had a smooth surface. It can be clearly seen that the size and morphology did not change significantly during the surface modification (Table 1 and Figure 1). The surface modification of SNPs was confirmed by FTIR spectroscopy; the results were previously published [35].

#### 2.1.2. GC Analysis of *Melaleuca Alternifolia* EO

The composition of MA was determined by gas chromatography. The components were identified by comparing their retention times and relative retention factors with standards and oils of known composition. Two parallel measurements were performed. The main compounds were *p*-cymene 35.2% and terpinene-4-ol 32.5%. A detailed composition is presented in Table 2.

#### 2.1.3. Characterization of PEs

Properties of PEs are influenced by many parameters, like the interfacial surface tension of the phases, size, wettability, and concentration of stabilizing particles, o/w phase ratio, and emulsification energy. In this study, we examined the influence of the o/w phase ratio and size of the stabilizing SNPs on the droplet size and stability of formulated PEs, while other parameters were kept constant.

The results can be seen in Table 3, including data for composition, droplet size, and appearance of PEs. For the microbiological experiments and in vitro diffusion studies, we used emulsions that were stable for at least one week, which means that their droplet size did not change within this period, and creaming, phase separation, aggregation, or sedimentation of SNPs did not occur.

As shown by Binks and Horozov [31], the size of stabilizing particles influences the emulsion droplet size at the same o/w phase ratio. We found that the increment of oil phase concentration caused an increment of emulsion droplet size at all examined oil phase concentrations. When 20 nm SNPs were used, the droplet size increased until reaching a droplet size of 1.8 µm. Further increment of oil phase concentration did not cause significant droplet size increment, and the stability of emulsions was much higher in the 11.16–16.12 mg/mL concentration range (for oil phase) (see Table 3). We observed a similar effect when 50 nm SNPs were used in the 4.48–11.19 mg/mL concentration range and 1.6 µm droplet size (Table 3). We could not observe such behavior for PE 100ET. In this case, the droplet size continuously increased as the oil phase concentration increased, and the stability of emulsions was much lower (less than one week).

The stability of PE was also influenced by the size of the stabilizing SNPs. We found that with the increasing SNPs size, the stability of PE decreased, at the same o/w ratio. The PE stability was 20 weeks using 20ET, 8 weeks for 50ET, and 1 week for 100ET SNPs, at 11.19 mg/mL oil phase concentration. The zeta potential values of PEs could also provide their colloidal stability, whose values did not differ significantly from the SNPs suspensions. The zeta potential of 20ET, 50ET, and 100ET SNPs was −28.3, −25.2, and −25.0 mV, while the values of PE 20ET, PE 50ET, and PE100Et in the 0.90–17.91 mg/mL oil concentration range were from −28.0 to −19.3 mV, −24.9 to −19.0 mV, and −24.3 to −17.6 mV, respectively.

The type of emulsions was determined by conductivity measurements. The conductivity values of stabilizing SNPs suspended in distilled water were 215.0, 211.4, and 268.3 μS∙cm^−1^ for 20ET, 50ET, and 100ET, respectively, while conductivity for the oil phase was 0.058 µS∙cm^−1^. The conductivity values of PEs were in the 157.33–257.50 µS∙cm^−1^ range, which means that all the PEs were o/w type emulsions.

#### 2.1.4. In Vitro Diffusion Studies through Artificial Membranes

Our goal was to formulate an emulsion that is capable of delivering the antifungal drugs through the nail plate and retain the drugs in the site of the infection (nail bed) for a prolonged time to provide a sustained drug release. The diffusion studies of PEs on the artificial membranes were performed in Franz diffusion vertical cells in order to examine the drug delivery ability of the formulated PEs. For diffusion studies, we applied PEs, CE, and ES of the same concentration, 17.91 mg/mL, as an antifungal drug combination. Because of the droplet size similarity between the CE and PEs, we can assume that only the dosage form determined the diffusion properties of the drug.

We found that PEs possessed better drug delivery properties through agar gel membrane compared to CE and ES (Table 4 and Figure 2). We examined the diffusion properties of PEs with different droplet sizes and found that the PEs with smaller droplet size (1.85 µm) could deliver as much as 89.9% of TIO through the agar membrane. In the experiment where the composite membrane was used, we found that the ES had diffused through the composite membrane structure, and only a small portion (2.4%) of the drug remained in the composite membrane (Table 4 and Figure 3). The PE 20ET delivered 89.9% of TIO through the agar gel membrane, and only 5.7% had diffused through composite membranes, suggesting that 84.2% of the applied drug remained in the targeted area. This amount was 61.1% at PE 50ET and 45.13% at PE 100ET. These in vitro experimental results suggested that PEs had better on-site drug delivery properties.

### 2.2. Microbiological Tests Using Candida albicans and Trichophyton rubrum

Data obtained for the minimum inhibitory concentration (MIC) and minimum fungicidal concentration (MFC) on *T. rubrum* and on *C. albicans* are shown in Table 5 and Figure 4 and Figure 5 for the ethanolic solution of TIO (ES-TIO) and ethanolic solution of MA (ES-MA) and their combinations. The TIO and MA combination showed a significant synergistic effect. When *T. rubrum* and *C. albicans* were treated with the combination of TIO and MA, both the MIC and MFC values decreased significantly compared to the separately used drugs.

Analyzing the antimicrobial data of the different formulations of TIO and MA clearly showed that the PEs were more effective than CE or ES against the two pathogens. The PE 100ET showed the most effective growth inhibition against both *T. rubrum* and *C. albicans*, and this formulation had the highest fungicidal activity.

## 3. Materials and Methods 

### 3.1. Preparation and Characterization of Silica Nanoparticle-Stabilized Pickering Emulsions

#### 3.1.1. Synthesis, Surface Modification, and Characterization of Silica Nanoparticles

Synthesis of hydrophilic SNPs (HS) was performed based on the work of Stöber et al. [36]. Size-selective synthesis parameters were set based on our previous work [37], as well as the reaction circumstances for surface modification. The synthesis route and details can be found in the Appendix A. We previously reported that SNPs that have theoretical surface coverage of 20% with ethyl groups could stabilize the MA droplets to give a stable PE [24]. For the synthesis and surface modification of silica nanoparticles, the following chemicals were used: tetraethoxysilane ([TEOS] (Alfa Aesar GmbH, Karlsruhe Germany, purity 98%), ethyltriethoxysilane ([ETES] Alfa Aesar Karlsruhe Germany, purity 96%), absolute ethanol (VWR Chemicals Ltd., Debrecen Hungary, AnalaR NORMAPUR^®^ ≥99.8%), and 28 m/m% ammonium solution (VWR Chemicals Ltd., Debrecen Hungary, AnalaR NORMAPUR^®^, analytical reagent).

The size distribution was determined by dynamic light scattering (DLS) (Malvern Zetasizer Nano S, Malvern Panalytical Ltd., Great Malvern, Worcestershire, UK). The size distribution was confirmed, and the morphology of silica nanoparticles was studied by transmission electron microscopy (TEM) (JEOL-1400 electron microscopy, JEOL Ltd., Tokyo, Japan). For TEM experiments, 400 mesh copper grids coated with carbon were used (Micro to Nano Ltd., Haarlem, Netherlands).

#### 3.1.2. Gas Chromatography Analysis of *Melaleuca Alternifolia* Essential Oil

##### Solid-Phase Microextraction (SPME) Conditions

Samples were loaded into vials (20 mL headspace) sealed with a silicon/PTFE septum prior to SPME-GC/MS analysis. Sample preparation using the static headspace solid-phase microextraction (sHS-SPME) technique was carried out with a CTC Combi PAL (CTC Analytics AG, Zwingen, Switzerland) automatic multipurpose sampler using a 65 μM StableFlex polydimethyl siloxane/carboxene/divinyl benzene (CAR/PDMS/DVB) SPME fiber (Supelco, Bellefonte, PA, USA). After an incubation period of 5 min at 100 °C, extraction was performed by exposing the fiber to the headspace of a 20 mL vial containing the sample for 10 min at 100 °C. The fiber was then immediately transferred to the injector port of the GC/MS and desorbed for 1 min at 250 °C, in split mode, and the split ratio was 1:90. The SPME fiber was cleaned and conditioned in a Fiber Bakeout Station in a pure nitrogen atmosphere at 250 °C for 15 min.

##### GC-MS Conditions

The analyses were carried out with an Agilent 6890N/5973N GC-MSD (Agilent Technologies, Santa Clara, CA, USA) system equipped with Supelco (Sigma-Aldrich Ltd., Budapest, Hungary) SLB-5MS capillary column (30 m × 250 µm × 0.25 µm). The GC oven temperature was programmed to increase from 60 °C (3 min isothermal) to 250 °C at 8°C/min (1 min isothermal). High purity helium (6.0) was used as carrier gas at 1.0 mL/min (37 cm/s) in constant flow mode. The mass selective detector (MSD) was equipped with a quadrupole mass analyzer and was operated in electron ionization mode at 70 eV in full scan mode (41–500 amu at 3.2 scan/s). The data were evaluated using MSD ChemStation D.02.00.275 software (Agilent Technologies, Santa Clara, CA, USA). The identification of the compounds was carried out by comparing retention data and the recorded spectra with the data of the NIST 2.0 library. The percentage evaluation was carried out by area normalization.

#### 3.1.3. Determination of Solubility of Tioconazole in *Melaleuca Alternifolia* Essential Oil

##### Solubility Calculations by Hansen Solubility Parameters (HSPs)

As a preliminary study, the calculations of solubility parameters were performed using the Hansen Solubility Parameters in Practice (HSPiP) software version 5.0.11 using the simplified molecular-input line-entry system (SMILES), obtained from PubChem. HSPs (Equation (1)) use group contribution to split the total cohesion energy of a solvent into contributions from atomic dispersion (*δ_d_*), polar interactions (*δ_p_*), and hydrogen bonding (*δ_h_*) [38].
(1)$$\delta ={\left(\delta_{d}^{2}+\delta_{p}^{2}+\delta_{h}^{2}\right)}^{0.5}$$

Differences in solubility parameters were calculated with the HSP difference (Equation (2)). A value below that of the reported cut-off value 7 Mpa^0.5^ indicates miscibility [39].
(2)$$\Delta \delta =\left|\delta_{solvent}-\delta_{tioconazole}\right|$$

For the calculation, the three main components of MA were used (*p*-cymene, terpinene-4-ol, γ-terpinene), and it could be established that TIO can be dissolved in the EO. The results of the calculation can be found in Appendix A. In order to determine the exact solubility of TIO in MA, the solvent addition method was performed (Section 3.1.3 Determination of Kinetic Solubility).

##### Determination of Kinetic Solubility

The kinetic solubility of TIO (tioconazole, purity ≥98%, Alfa Aesar, Karlsruhe, Germany) in water-saturated MA (*Melaleuca alternifolia* essential oil, Tebamol^®^, BIO-DIÄT-BERLIN GmbH, Berlin, Germany) was determined by the solvent addition method [40]. The examination was performed at ambient temperature (25 °C). The initial suspension was prepared by weighing the exact amount of 1.0 mg TIO and the addition of 500 µL of MA. The volume of MA was increased until the suspension turned into a clear solution. The light scattering of suspension was determined visually and with instrumental measurement of scattered light intensity (DLS, Malvern Zetasizer Nano S). The kinetic solubility of TIO in MA was found to be 0.213 mg/mL (23.8 *m/m*%). The concentration of TIO in MA was set to 20 *m/m*% for the PE preparations.

#### 3.1.4. Preparation and Characterization of Pickering Emulsions

The concentration of emulsifiers in the water phase was set to 1 mg/mL and was kept constant for all experiments. Three different sizes of SNPs were used for PE formulation (20ET, 50ET, 100ET) and Tween80^®^ surfactant (Tw80) (Polysorbate80 Acros Organics, Thermo Fisher Scientific, Waltham, MA, USA) for conventional emulsions (CE). The concentration of oil phase varied between 0.90 and 17.91 mg/mL, and the ratio of TIO and MA was always constant (20 *m/m*%).

The emulsification was performed in two steps. The coarse emulsions were prepared by sonication for 2 min (Bandelin Sonorex RK 52H, BANDELIN electronic GmbH & Co. KG, Berlin, Germany). The final emulsification was performed with UltraTurrax (IKA Werke T-25 basic, IKA Werke GmbH, Staufen im Breisgau, Germany) for 2 min at 13,500 rpm. The emulsions’ droplet size was determined with DLS using a Malvern Zetaziser Nano S instrument (Malvern Panalytical Ltd., Great Malvern, Worcestershire, UK). The stability of the emulsions was determined from periodical droplet size determination. The emulsions were stored at room temperature (25 °C).

The type of emulsions was determined with conductivity test using Mettler Toledo Seven2Go S3 conductivity meter (Mettler Toledo GmbH, Giessen, Germany) and InLab^®^ 738-ISM sensor (Mettler Toledo GmbH, Giessen, Germany).

All experiments, measurements, and standard deviation calculations were performed from 3 parallel sample preparations.

#### 3.1.5. In Vitro Diffusion Studies—Static Franz Diffusion Cell Method

Accepted models for testing drugs and their formulations for onychomycosis treatments include penetration tests through cadaver nails [41], nail clippings, bovine hoof slices, or keratin films [42] made from human keratin source. The non-uniformity of natural membranes causes huge inhomogeneity in the results [43,44,45], which makes the comparison of different formulations impossible. The aim of our study was to examine the diffusion properties of applied drugs in complex colloidal systems; therefore, in our opinion, the similarity in hydrophilicity and surface charge between the nail plate or nail bed and artificial membranes was of the highest importance for testing and comparison of the formulations. The nail plate acts as a negatively charged aqueous hydrogel, as it is described in the literature [46], and it has properties similar to that of the agar gel [47]. Based on the literature data obtained from independent researches, we compared the diffusion coefficient and flux of well-studied antibiotic chloramphenicol (5 mg/mL in phosphate-buffered saline) with different membranes, namely agar gel [48], bovine hoof slice, and cadaver nail plate [49]. We found that the diffusion coefficients and flux values were very similar for agar gel and bovine hoof slice membranes. Flux for bovine hoof was 4.07 ± 1.18∙10^−6^ mg/cm^2^∙s, for agar gel 1.96 ± 0.47 10^−6^ mg/cm^2^∙s, and 8.21 ± 2.11 10^−7^ mg/cm^2^∙s for the cadaver nail plate. The flux values for agar gel were closer to the value for the cadaver nail plate, which might suggest that agar gel is a good model membrane for water-based formulations.

The agar gel membrane was used to model the nail plate. The composite membrane, consisting of the agar gel layer on top of the cellulose acetate membrane, was used to simulate the complex structure of nail plate and nail bed since nail bed has similar properties as skin [50,51], and the cellulose membrane has been commonly used as a model for skin permeability [52]. The main aim of the study on two types of membranes was to examine whether the examined formulations could deliver the applied lipophilic drugs through agar gel as a model for nail plate, and the composite membrane was used to examine if the formulation could retain the drugs on the main site of the infection, namely nail bed. The amount of the drug transported through the membrane was calculated based on the amount introduced to the membrane. In the case of agar membrane, the goal was to prepare the drug delivery system that could deliver the highest drug amount through that membrane. The composite membrane was used to test the on-site retention of drugs in different formulations. The amount of retained drug was calculated as a difference between the drug amount passed through the agar gel membrane and the amount passed through the composite membrane

For in vitro testing, the 2.1 mm thick 6 *m*/*m*% agar gel membrane (Agar powder, purity >95%, VWR Chemicals Ltd., Debrecen, Hungary) and the same agar gel membrane combined with 0.8 mm thick cellulose acetate with effective penetration area of 2.54 cm^2^ (Membranfilter Porafil, Macherey-Nagel GmbH&Co. KG, Düren, Germany, pore size 0.2 µm) were used. Before each measurement, the agar gel was always freshly prepared. The agar powder was dispersed in demineralized water, and the mixture was boiled in a closed vial for 3 min until all agar was completely dissolved. Exactly 10 mL of agar gel was poured into a plastic vessel (i.d. 70.8 mm), then left to cool (25 °C) and gelate. After the gelation, the agar gel was soaked in PBS buffer for 12 h. Finally, the agar membrane was cut out with a sharp home-made tool and placed on the Franz cell. The cellulose acetate membranes were also freshly soaked in PBS buffer before the experiments.

The examination of diffusion properties was performed at 32 °C in static vertical Franz diffusion cells (Hanson Microette Plus, Hanson Research 60-301-106, Hanson Research Corporation, Chatsworth, CA, USA); six parallel cells with effective penetration area 2.54 cm^2^ were used, and each experiment was made in triplicates. The volume of the receiver chamber was 7 mL; the receiver solution was PBS buffer. For PBS preparation, the following salts were used: NaCl (high purity, VWR Chemicals Ltd., Debrecen Hungary), KCl (purity 99–100.5%, VWR Chemicals Ltd., Debrecen Hungary), Na_2_HPO_4_∙2H_2_O (AnalaR NORMAPUR^®^, purity ≥99.0%, VWR Chemicals Ltd., Debrecen Hungary), and KH_2_PO_4_ (purity ≥99.0%, VWR Chemicals Ltd., Debrecen Hungary). The 600 µL volume of emulsion or solution sample was placed into the donor chamber, and the diffusion was examined for 2 h; the stirring rate was 700 min^−1^, and the samples were collected after 5, 10, 15, 30, 60, 90, and 120 min. The withdrawn sample volume was replaced with a fresh PBS buffer.

The TIO content was determined with HPLC measurements using UV-Vis detector (SPD 10-A, Shimadzu Europa GmbH, Duisburg, Germany); the method is based on Bagary et al. [53]. Separations were carried out using a monolithic silica type column (ODS-AM302, S-5μm, 120A, YMC Co., Kyoto, Japan). The mobile phase consisted of methanol/0.02 M K_2_HPO_4_ = 85/15 *V*/*V*% and 0.2 *V*/*V*% trimethylamine (methanol dehydrated, ultrapure ≥99.8%, VWR Chemicals Ltd., Debrecen Hungary; trimethylamine: HiPerSolv CHROMANORM^®^, VWR Chemicals Ltd., Debrecen Hungary), pH = 7.0. The mobile phase was freshly filtered through Millipore Nylon membrane (pore size: 0.2 µm, Merck KGaA, Darmstadt, Germany) before the analysis. Isocratic elution was programmed with a 1.5 mL/min flow rate; the temperature of measurement was 32 °C. The detection wavelength of tioconazole was 254 nm, and its retention time was 3.5 min.

### 3.2. Microbiological Tests against Candida albicans and Trichophyton rubrum

#### 3.2.1. Instruments Used in the Microbiological Experiments

UV-Vis spectrophotometer (Hitachi U-3900, Hitachi High-Tech Corporation, Japan), microbiological incubator (Thermo Scientific Heraeus B12, Thermo Fischer Scientific, Waltham, MA, USA), Bürker cell counting chamber (Hirschmann Laborgeräte GmbH & Co., Germany), Multiskan EX 355 (Thermo Fischer Scientific, Waltham, MA, USA) spectrophotometer were used throughout the experiments.

#### 3.2.2. Materials Used in the Microbiological Experiments

For the microbiological experiments, the following materials were used: sterile 96-well microtiter plates (Greiner Bio-One, Kremsmunster, Austria), potato dextrose agar (PDA) (BioLab, Budapest, Hungary), sterile filter inserts (pore size 10 µm) from PluriSelect (pluriSelect Life Science, Leipzig, Germany), dextrose, adenine, bacteriological peptone and agar-agar (Reanal Labor, Budapest, Hungary), sterile centrifuge tubes (TPP Techno Plastic Products, Trasadingen, Switzerland), homemade Sabouraud dextrose agar or SDA (containing 4% dextrose, 1% bacteriological peptone, and 1.5% agar-agar in double-distilled water), yeast extract peptone dextrose agar (containing 2% bacteriological peptone, 1% yeast extract, 2% dextrose, and 1.5% agar-agar in double-distilled water), 3-(*N*-Morpholino)-propanesulfonic acid (MOPS) from Serva Electrophoresis GmBH (Heidelberg, Germany), and RPMI 1640 medium (containing 3.4% MOPS, 1.8% dextrose, and 0.002% adenine) from Sigma-Aldrich Chemie GmBH (Steinheim, Germany). Highly purified water (˂1.0 µS) was applied throughout the experiments.

#### 3.2.3. Fungal Cultures and Inoculum Preparation

*Trichophyton rubrum* (*T. rubrum*) DSM 21146 and *Candida albicans* (*C. albicans*) ATCC 001 were obtained from Leibniz Institute DSMZ GmbH (Braunschweig, Germany) and from Department of General and Environmental Microbiology (Institute of Biology, University of Pécs, Hungary), respectively.

We followed the methods described previously [54,55,56,57] for *T. rubrum* and *C. albicans* culture preparation. In brief, *T. rubrum* stock inoculum suspensions were prepared from 7-day old cultures grown on PDA at 28 °C for sporulation. Ten days later, the observed fungal colonies were flooded with 10 mL distilled water, followed by scraping the surface using a sterile loop. Conidia and hyphal mixed suspensions were withdrawn and were transferred to a sterile centrifuge tube through sterile filter inserts (10 μm, pluriSelect) to remove hyphae, leaving a filtered inoculum containing spores only. The inoculum cell population was adjusted to 0.5 to 5 × 10^6^ spores/mL visually using a Bürker cell counting chamber, followed by further turbidity calibration with a UV-Vis spectrophotometer (Hitachi U-3900) at 520 nm. The spores were further diluted to the desired population according to the experimental requirements.

*C. albicans* stock inoculum was prepared from 48 h old culture grown on YEPD agar plates at 30 °C. After 18 h of incubation at 30 °C in a microbiological incubator, on YEPD agar slant, the cells were looped out, diluted with 0.9% sterile saline, and were counted by a Bürker cell counting chamber, followed by turbidity calibration with a UV-Vis spectrophotometer (Hitachi U-3900) at 595 nm. The fungal cell population was set to ~1 × 10^6^ cells/mL and was diluted later according to the experimental designs.

#### 3.2.4. Determination of Antifungal Activities

For the evaluation of the minimum inhibitory concentration (MIC) of *T. rubrum* and *C. albicans*, we followed previously published methods [56,57,58,59]. The ethanolic solutions of TIO and MA in a wide concentration range (0.5–300 µg/mL) were used for the assay. CE and PEs formulations were also tested; an initial concentration of the oil phase was 160 µg/mL for *T. rubrum*, whereas 180 µg/mL for *C. albicans* treatment was applied. The treating mixtures were further diluted up to 256 times in a serial half-dilution format.

One hundred microliters of fungal cell suspensions (see Section 3.2.5 and Section 3.2.6) with equal fungal contents were applied thereafter to the microplate wells containing 100 µL of the different samples. Detailed information on the assay conditions can be found in Section 3.2.5 and Section 3.2.6. As a blank, suspensions of 20ET, 50ET, 100ET SNPs, pure ethanol, Tw80 solution were used.

#### 3.2.5. Determination of Minimum Inhibitory Concentration of *T. rubrum*

The *T. rubrum,* inoculum size of ~2.5 × 10^4^ spores/mL, containing the test drugs in half-dilution format, was incubated in RPMI media for 7 days in a microbiological incubator at 28 °C. The microplates containing *T. rubrum* incubated for 7 days with the test drugs were evaluated following the protocol as recommended by the Clinical and Laboratory Standards Institute (CLSI) M38-A2. The untreated cell samples and the medium without cells were considered as the growth control and blank, respectively. The endpoint determination readings for the minimum inhibitory concentrations (MIC) were performed visually based on the comparison of the growth in the wells containing the test drugs with that of the growth control [60]. All evaluations were performed in triplicates in six independent experiments.

#### 3.2.6. Determination of Minimum Inhibitory Concentration of *C. albicans*

A population size of ~2 × 10^3^ CFU/mL was incubated in RPMI media with the above-mentioned test drug concentration range at 30 °C for 48 h in the case of *C. albicans*. A Multiskan EX 355 spectrophotometer was used to measure the absorbance (at 595 nm) of the samples in the microtiter plate in the case of *C. albicans*. The absorbance values of the respective treatments were converted to a percentage and were compared to growth control (100%). The untreated fungal samples and the medium without cells were considered as the growth control and blank, respectively. All evaluations were performed in triplicates in six independent experiments.

#### 3.2.7. Determination of the Minimum Fungicidal Concentration (MFC)

Determination of MFC was performed using the methods as described earlier with modifications [56]. After performing the MIC, 10 µL of the content from each well (not visibly turbid) was inoculated onto sterile SDA plates. The plates were incubated at 30 °C for 48 h. MFC was evaluated as the lowest drug concentration, resulting in no growth (≥99.9% growth inhibition). Measurements were performed by applying three technical replicates in six independent experiments.

#### 3.2.8. Statistical Analyses

The statistical analyses were conducted using a one-way ANOVA test (Origin 2016, OriginLab Corp., Northampton, MA, USA), and the significance was set at *p* ≤ 0.05.

## 4. Conclusions

The choice of drugs used in this research was based on careful consideration. The TIO was chosen as a drug with high antifungal activity but low water solubility and permeability through the nail plate. MA EO was selected because of the antifungal activity and because it is a liquid and can be used as a solvent for TIO. The combination of the drugs applied in this study showed a significant synergistic effect. The solution of TIO in MA EO was successfully formulated into stable Pickering emulsions. In vitro studies have demonstrated that PEs are effective drug formulations that can provide site-specific and effective drug delivery through artificial membranes. 20ET PE achieved the highest drug delivery efficiency as it could deliver 40% of the drug introduced to the artificial membrane within 10 min. The amount delivered at this time was 572 µg of TIO through the agar model membrane, while the MFC of the TIO in this formulation was 4.69 µg/mL. To prove the real applicability of the suggested drug combination and PE formulation, we have to perform experiments on the natural nail model. Still, from the presented data, we can conclude that the application of both site-specific drug delivery and synergistic antifungal drug combinations is a promising route for the development of effective onychomycosis topical treatment formulation.

## Figures and Tables

**Figure 1 molecules-25-05544-f001:**
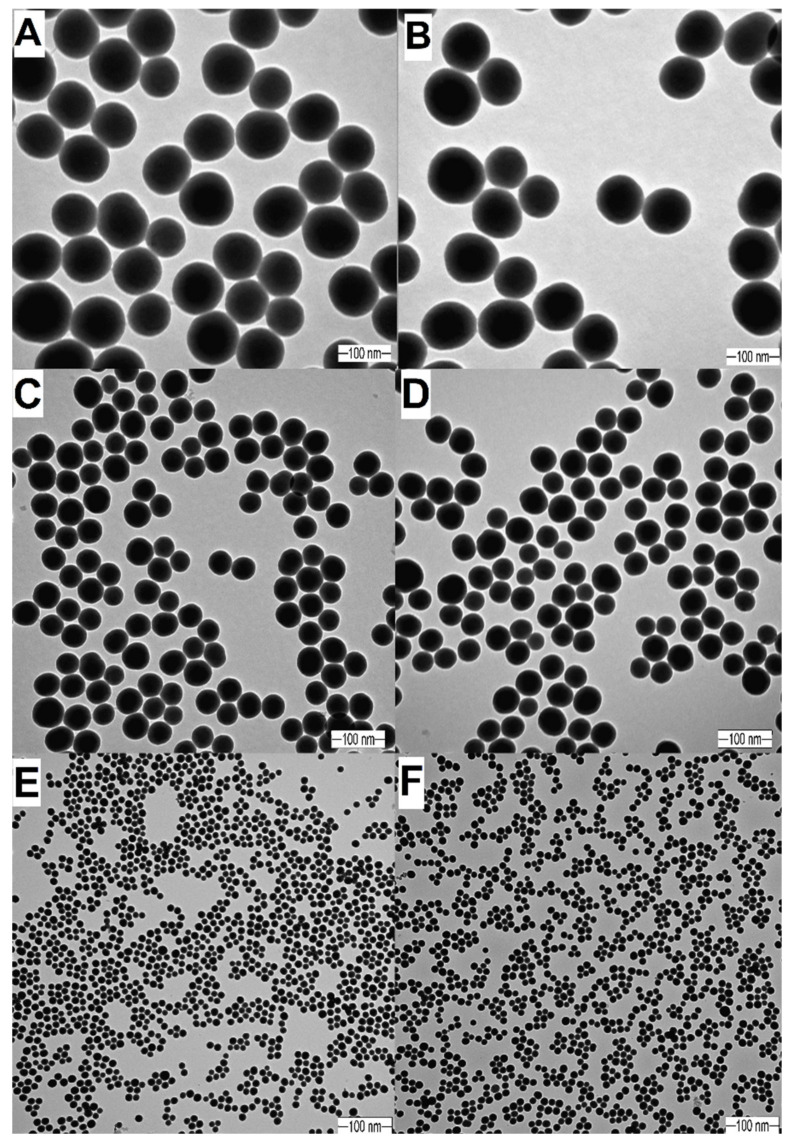
TEM images of silica nanoparticles (SNPs). (**A**): HS100, d = 103.0 nm. (**B**): ET100, d = 110.7 nm. (**C**): HS50, d = 53.0 nm. (**D**): ET50, d = 55.0 nm. (**E**): HS20, d = 20.0 nm. (**F**): ET20, d = 20.0 nm.

**Figure 2 molecules-25-05544-f002:**
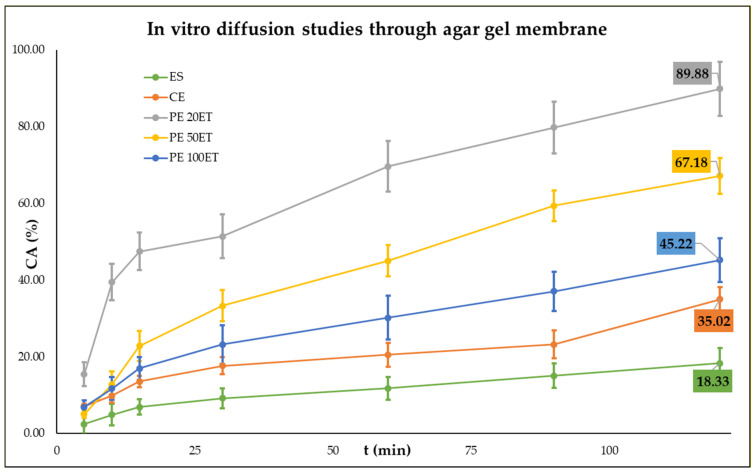
In vitro diffusion studies through agar gel membrane. ES: ethanolic solution, CE: conventional emulsion, PE: Pickering emulsion, CA: cumulative TIO amount after 2 h. C_TIO_ = 3.58 mg/mL.

**Figure 3 molecules-25-05544-f003:**
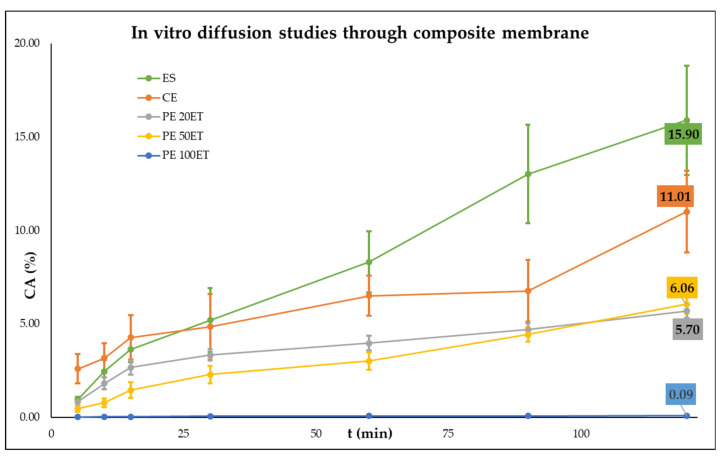
In vitro diffusion studies through the composite membrane. ES: ethanolic solution, CE: conventional emulsion, PE: Pickering emulsion, CA: cumulative TIO amount after 2 h. C_TIO_ = 3.58 mg/mL.

**Figure 4 molecules-25-05544-f004:**
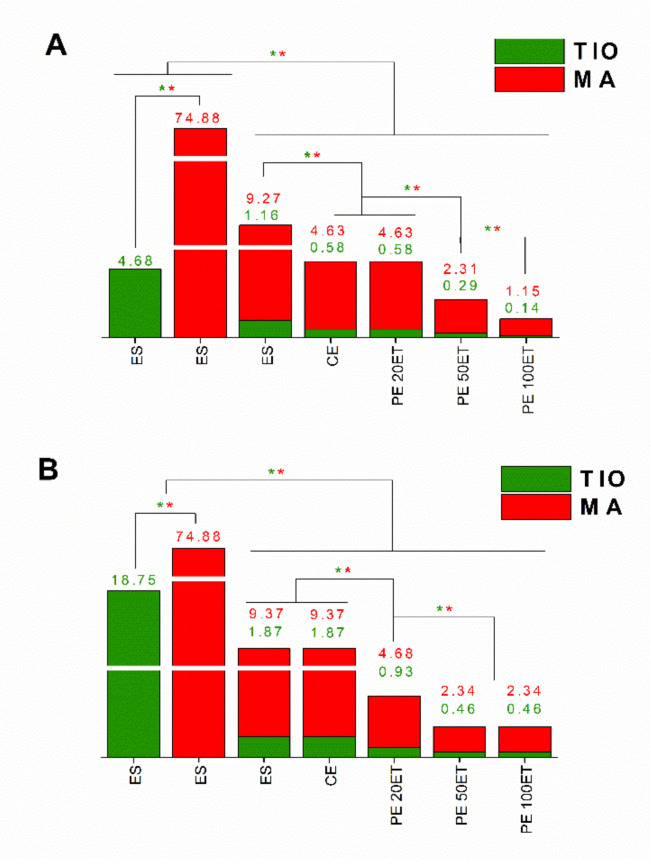
Minimum inhibitory concentration (MIC_90_) of ES-TIO, ES-MA, ES-TIO-MA, CE-TIO-MA, PE 20ET-TIO-MA, PE 50ET-TIO-MA, and PE 100ET-TIO-MA in μg/mL on *T. rubrum* (**A**) and *C. albicans* (**B**). Six independent experiments with three technical replicates were performed. The green (*****) and red (*****) asterisks represent a significance value of *p* ˂ 0.01 for the MIC_90_, respectively.

**Figure 5 molecules-25-05544-f005:**
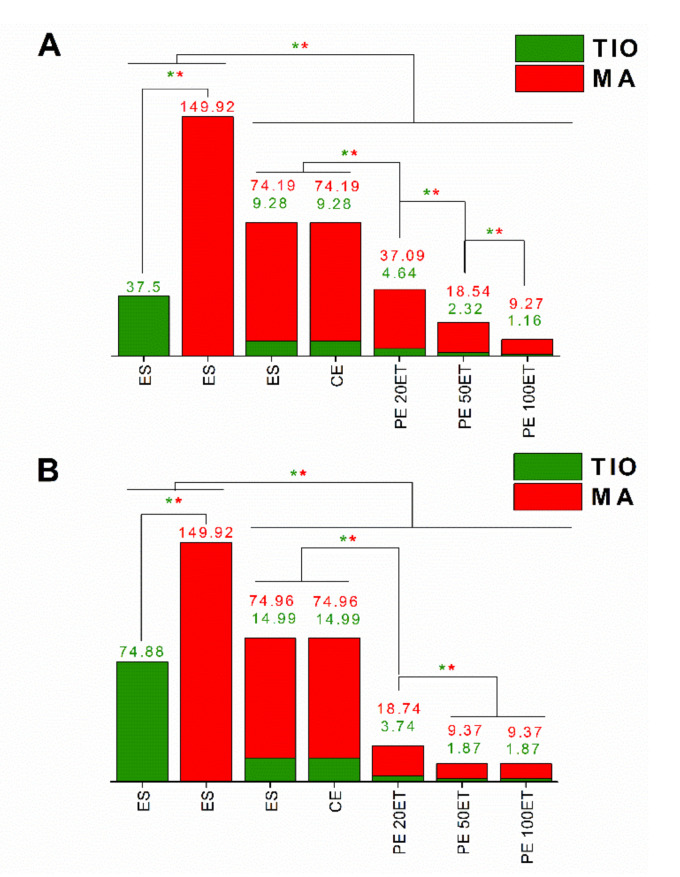
Minimum fungicidal concentration (MFC) of ES-TIO, ES-MA, ES-TIO-MA, CE-TIO-MA, PE 20ET-TIO-MA, PE 50ET-TIO-MA, and PE 100ET-TIO-MA in µg/mL on *T. rubrum* (**A**) and *C. albicans* (**B**). Six independent experiments with three technical replicates were performed. The green (*****) and red (*****) asterisks represent a significance value of *p* ˂ 0.01 for the MFC, respectively.

**Table 1 molecules-25-05544-t001:** Properties of SNPs. HS: hydrophilic particle. ET: with ethyl functional group modified particle. Size represented as mean ± SD of three parallel syntheses. The numbers refer to the particle sizes.

Samples	d_DLS_ (nm)	PDI_DLS_	d_TEM_ (nm)	PDI_TEM_
20HS	20.1 ± 0.2	0.008	20.0	0.011
50HS	52.7 ± 0.9	0.017	53.0	0.037
100HS	105.2 ± 3.6	0.021	103.0	0.083
20ET	20.1 ± 0.8	0.158	22.0	0.210
50ET	54.2 ± 2.7	0.178	55.0	0.337
100ET	110.7 ± 4.1	0.231	112.0	0.349

**Table 2 molecules-25-05544-t002:** Composition of *Melaleuca alternifolia* EO. The results of GC analysis showed the average of the two parallel measurements in the percentage of volatile compounds. The main components of MA have been highlighted.

Compounds	Retention Time t_R_ (min)	Percentage Ratio of Compounds (%)
α-thujene	5	1.7
α-pinene	5.2	4.6
β-phellandrene	5.2	0.6
β-pinene	6.2	1.2
β-mycrene	6.4	0.8
α-terpinene	7.0	1.4
***p*-cymene**	**7.3**	**35.2**
terpinyl-acetate	7.3	2.1
cineole	7.4	5.8
γ-terpinene	8	7.6
terpinolene	8.6	1.7
**terpinene-4-ol**	**10.7**	**32.5**
α-terpineol	11	2.6
aromadendrene	15.5	0.7
epiglobulol	16.4	1.2

**Table 3 molecules-25-05544-t003:** Parameters of Pickering emulsions of *Melaleuca alternifolia* EO and tioconazole stabilized with 20ET, 50ET, and 100ET SNPs. * These emulsions were creaming, but after 30 s shaking, their droplet size recovered to the original value, and they retained again their stability for 1 week.

	PE 20ET		PE 50ET		PE 100ET	
Coil Phase (mg/mL)	Droplet Size (nm)	Stability	Droplet Size (nm)	Stability	Droplet Size (nm)	Stability
0.90	4306 ± 39.6	10 min	1280 ± 62.8	1 day	1070 ± 438.5	30 min
1.79	615 ± 22.2	30 min	1320 ± 95.9	1 day	1350 ± 531.8	30 min
2.69	890 ± 103.3	30 min	1440 ± 83.5	1 day	1630 ± 464.5	30 min
3.58	1250 ± 94.5	1 day	1650 ± 51.5	2 day	1730 ± 514.5	10 min
4.48	1320 ± 32.5	8 weeks	1670 ± 216.8	2 day	1850 ± 107.9	10 min
5.37	1440 ± 100.2	8 weeks	1620 ± 79.7	8 weeks	1890 ± 333.8	10 min
6.27	1470 ± 35.2	8 weeks	1610 ± 34.4	8 weeks	1950 ± 95.0	10 min
7.16	1470 ± 62.5	8 weeks	1670 ± 62.8	8 weeks	1940 ± 20.1	1 week *
8.96	1660 ± 56.7	8 weeks	1690 ± 70.4	8 weeks	2070 ± 51.2	1 week *
11.19	1890 ± 41.2	20 weeks	2200 ± 188.9	8 weeks	2200 ± 59.5	1 week *
13.43	1840 ± 141.0	20 weeks	2250 ± 170.8	2 weeks *	2800.0 ± 85.7	1 week
16.12	1820.0 ± 99.6	20 weeks	2080 ± 160.1	2 weeks *	2850 ± 184.3	1 week
17.91	1850 ± 496.6	8 weeks	2380 ± 157.0	2 weeks *	3090 ± 116.6	1 week
**Coil Phase (mg/mL)**	**Appearance** **PET 20ET**		**Appearance** **PET 50ET**		**Appearance** **PET 100ET**	
0.90	creaming		sedimentation		creaming	
1.79	creaming		sedimentation		creaming	
2.69	creaming		sedimentation		creaming	
3.58	opalescent		opalescent		creaming	
4.48	opalescent		opalescent		creaming	
5.37	opalescent		milky		creaming	
6.27	opalescent		milky		creaming	
7.16	opalescent		milky		milky	
8.96	opalescent		milky		milky	
11.19	milky		milky		milky	
13.43	milky		milky		aggregation, sedimentation	
16.12	milky		milky		aggregation, sedimentation	
17.91	opalescent		milky		aggregation, sedimentation	

**Table 4 molecules-25-05544-t004:** Results of in vitro diffusion studies. ES: ethanolic solution, CE: conventional emulsion, PE: Pickering emulsion, CA: cumulative TIO amount after 2 h. The concentration of TIO was 3.58 mg/mL in each formulation.

Samples	Stabilizing Agent	Droplet Size (nm)	CA Agar Gel (%)	CA Agar Gel (mg/cm^2^)	CA Composite Membrane (%)	CA Composite Membrane (mg/cm^2^)
ES	-	-	18.33	0.26	15.90	0.22
CE	Tween80	2470.0 ± 89.1	35.02	0.49	11.01	0.15
PE 20ET	20ET SNPs	1850 ± 496.6	89.88	1.26	5.70	0.08
PE 50ET	50ET SNPs	2380 ± 157.0	67.18	0.95	6.06	0.05
PE 100ET	100ET SNPs	3090 ± 116.6	45.22	0.63	0.09	0.001

**Table 5 molecules-25-05544-t005:** Minimum inhibitory concentration (MIC) and minimum fungicidal concentration (MFC) of the test samples in combinations on *T. rubrum* and on *C. albicans.*

Sample	*T. rubrum*		*C. albicans*	
	MIC (µg/mL)	MFC (µg/mL)	MIC (µg/mL)	MFC (µg/mL)
ES-TIO	4.68	37.5	18.75	74.88
ES-MA	74.88	149.92	74.88	149.92
ES-TIO-MA	10.43	83.47	11.24	89.95
CE	5.21	83.47	11.24	89.95
PE 20ET	5.21	41.73	5.61	22.48
PE 50ET	2.6	20.86	2.8	11.24
PE 100ET	1.29	10.43	2.8	11.24

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
