# Peer review of "Formulation of Tioconazole and Melaleuca alternifolia Essential Oil Pickering Emulsions for Onychomycosis Topical Treatment"

_molecules, 2020, doi:10.3390/molecules25235544_

Round 1
Reviewer 1 Report
The novelty of this research is not highlighted in the manuscript. Neither Pickering Emulsions as drug delivery systems nor drugs investigated are new. Perhaps the only novelty is the use of Melaleuca alternifolia (MA) EO (tea tree EO) as a solvent for TIO to achieve a synergistic effect of the drugs. Unfortunately, the authors did not mention it.
Reviewer 2 Report
In my opinion, the paper here presented is very interesting and could be published in Molecules, but there are some aspects that should be considered prior its publication. Please, see specific comments.
SPECIFIC COMMENTS:
English styleshould be revised throughout the text. It is sometimes difficult to follow the meaning of some sentences. Let’s take an example from the abstract: ‘For stabilization of PEs silica nanoparticles with tailored size and partially hydrophobic surface have been synthesized’. Please, refine the redaction of this sentence and revise the whole manuscript.
General structure of the manuscript:
It would be clearer to describe Materials and Methods as point 2 and then, Results and discussion as point 3. On needs to go ahead in the manuscript to understand how the experiments were made when analyzing the results and then go back to follow the discussion. Please, change the order of both chapters.
Page 12 line 302 - In vitro diffusion studies:
It is not clear how the agar membranes were prepared. Please, describe the method for their elaboration and how did the authors control the thickness of the membranes
Reviewer 3 Report
The authors describe to prepare pickering emulsions including an antifungal drug, ticonazole and essential oil for onychomycosis topical treatment. This pickering emulsion system enhanced the membrane permeability and antifungal effect. This reviewer concludes that the result is interesting for readers of Molecules. However, this reviewer thinks that the results are not well discussed. The result and discussion part must be improved with more explanations and discussions, so this reviewer do notaccept the current version of manuscript for publish to the Molecules. This reviewer raises the following concerns.
1) The authors have already reported the characterization effect of pickering emulsions including essential oils. In this paper, the authors described the parameters of pickering emulsions of essential oil and drug stabilized with silica nanoparticle in Table 3, 4 and 5. This reviewer checks these parameters comparing the previous reports, the values are almost same. Please explain why these parameters of pickering emulsions do not change despite the presence of the drug in emulsion.
2) The main constituents of the essential oil are p-cymene and terppinene-4-ol, which together account for more than 70% of the total, but don't these two constituents alone show antifungal activity? or do the authors have any information of the antifungal activity of p-cymene and/or terppinene-4-ol? Please try to discuss this issue.
3) Table 3 to 5 should be combined into one.
4) Zeta potential values of Pickering emulsion should be added in the manuscript. Zeta potential is very important for stability and nail plate permeation.
5) There is no reason why the changing the size of silica nanoparticle affect the stabilization's term of pickering emulsions. Small size silica nanoparticles seem to prolong the stability of pickering emulsions.
5) The authors described the agar gel membrane is used for permeation study instead of cadaver nail plate, but I feel this method is questionable, because I did not find any report to use the agar membrane for drug permeation through the nail instead of nail plate in data base. Of course, this reviewer understand the authors do effort to reveal the correlation of drug permeation between nail plate and agar gel membrane using Chloramphenicol. But, the characterization of Chloramphenicol is different from tioconazole, the authors validated permeation correlation using only one drug, the agar gel might have the amount of water rather than nail plate and so on.. I believe this method is not supported clearly enough. Do you have any reference of agar gel for drug nail permeation? This needs to be addressed as well using the other result and/or information.
6) Why did the authors carry out the drug permeation studies of agar gel membrane and composite membrane? There is no explanation of why the two different membranes were tested.
7) Regarding to 6), the rank order of drug permeation of pickering emulsions tested in this study are different between agar gel membrane and composite membrane, the authors should describe why this difference is occurred.
8) The authors evaluated the amount drug permeation through both membrane and remained in both membrane by PE. Comparing the both membrane, the remained drug was higher than drug permeation through both membrane, but drug remained in composite membrane was up to 6% against much more 45% drug remained in agar gel. Try to discuss about this difference.
9) Also, the authors did not explain why PE100ET shows the minimum MIC and MFC value rather than not only CE but PE20ET and 50ET.
10) Refs. Check over references for formatting issues – for example be sure to use standard journal abbreviations instead of listing such as ref. #4 Br. J. Dermatrology → Br. J. Dermatol. There are a lot of spelling and abbreviation errors.
Round 2
Reviewer 3 Report
This reviewer carefully read through the revised manuscript, but this reviewer do not accept the current version of manuscript for publish to the Molecules.
The authors described that the main goal was not to determine the absolute value of drug permeability, but to compare the drug delivery ability of different formulations of the same drugs, but this reviewer can not agree with the author on this point.
The authors have evaluated the nail permeability of drugs and essential oils using agar gels, but even if the purpose is to evaluate the release from the formulation, this is not a reason to actively use agar gels.If the sole purpose is to evaluate the drug release from the PEs, there is no need to perform a drug permeability study through agar gel. Even if the permeability of chloramphenicol is similar between nails and agar gels, as we pointed out before, it is doubtful that agar gels are suitable as a model for evaluating nail permeability based on the similarity in permeability of only one drug. The reviewers understand it is difficult to carry out the permeability tests using animals and humans within the EU countries, however, other researchers have already used keratin sheets and other materials as alternative models of nail permiability (e.g., e.g., Christel C. Müller-Goymann, 2011, Eur. J. Pharm. Sci., https://doi.org/10.1016/j.ejpb.2011.01.022). Especially in agar gels, drug diffusion from hydrophilic emulsions is considered to be very fast because they contain a very large amount of water. In contrast, because the water content of nails is low, the evaluation of permeability using agar gel is not considered to be applicable to nail permeability. The title of this manuscript includes "Essential Oil Pickering Emulsions for Onychomycosis Topical Treatment", the authors should select appropriate permeation tests based on the target site.
My main point of emphasis is the result and discussion part must be improved with more explanations and discussions. The authors have not described the extended discussion pointed by reviewer in this manuscript and this reviewer feel the style of the writing is still similar more and more to a journalistic report, finally, this reviewer recommend the authors should rewrite this manuscript with and submit to the other journal(s).